# Brainstem networks construct threat probability and prediction error from neuronal building blocks

Jasmin A. Strickland [1,2] ✉ & Michael A. McDannald [1] ✉

When faced with potential threat we must estimate its probability, respond advantageously, and leverage experience to update future estimates. Threat estimation is the proposed domain of the forebrain, while behaviour is elicited by the brainstem. Yet, the brainstem is also a source of prediction error, a learning signal to acquire and update threat estimates. Neuropixels probes allowed us to record single-unit activity across a 21-region brainstem axis in rats receiving probabilistic fear discrimination with foot shock outcome. Against a backdrop of diffuse behaviour signaling, a brainstem network with a dorsal hub signaled threat probability. Neuronal function remapping during the outcome period gave rise to brainstem networks signaling prediction error and shock on multiple timescales. The results reveal brainstem networks construct threat probability, behaviour, and prediction error signals from neuronal building blocks.

Faced with a potential threat, we must estimate its probability, determine an appropriate response, and—should we come away intact—adjust our estimates for future encounters. Historical and current descriptions of the brain's threat circuitry emphasize a division of labor in which forebrain regions estimate threat probability, while the brainstem elicits behavior[1,2]. However, behavior signaling is not observed in expected brainstem neuronal populations, such as the periaqueductal gray[3,4], which instead signals threat probability[5]. Instead, periaqueductal behavior signaling is observed in an unexpected, cue-inhibited population[6]. Furthermore, the brainstem periaqueductal gray is a source of prediction error[3,7,8], a learning signal to adjust threat estimates[9]. These findings necessitate a more complex role for the brainstem in threat estimation. However, evidence of widespread brainstem threat probability signaling remains elusive, and more complete descriptions of brainstem behavior and prediction error signaling are needed. Recording a 21-region axis with Neuropixels[10] during probabilistic fear discrimination[11], we report the brainstem constructs signals for threat probability, fear behavior, and prediction error from neuronal "building blocks" organized into functional networks. Remapping of neuronal function between cue and outcome periods revealed distinct brainstem network organization for threat probability, fear behavior, and prediction error signaling.

## Results

Ten rats (four females) were first shaped to nose poke for food reward. Independent of poke-food contingencies, rats received probabilistic fear discrimination during which three cues predicted unique foot shock probabilities: danger ($p = 1$), uncertainty ($p = 0.25$), and safety ($p = 0$) (Fig. 1a). A 0.25 uncertainty probability was chosen because higher probabilities can produce behavior equivalent to danger[12]. Rats were implanted with a Neuropixels probe through the brainstem (Fig. 1b) to permit high-density, single-unit recordings from a complete dorsal-ventral axis during discrimination. Fear was calculated with a suppression ratio (see methods), comparing reward-seeking rates during baseline and cue periods. Ratio extremes indicated complete suppression of reward seeking (1) versus no suppression (0). Ratios between 0 and 1 indicated intermediate levels of suppression. Rats showed complete discrimination during recording sessions. Suppression of rewarded nose poking was high to danger, intermediate to uncertainty, and low to safety (Fig. 1c; ANOVA main effect of cue [$F_{(2,142)} = 149.2$, $p = 1.26 \times 10^{-35}$], Supplementary Fig. S2). We isolated

[1]Department of Psychology & Neuroscience, Boston College, Chestnut Hill, MA 02467, USA. [2]Department of Psychology, Durham University, Durham DH1 3LE, UK. ✉e-mail: jasmin.strickland@bc.edu; michael.mcdannald@bc.edu

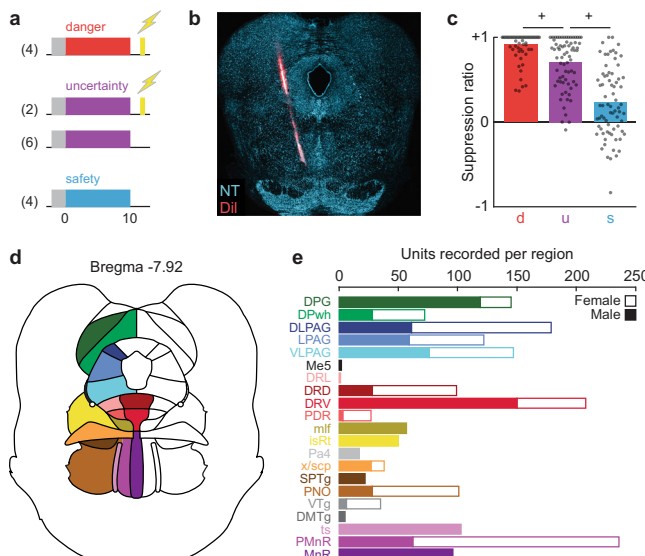

**Fig. 1 | Fear discrimination and neuropixels implant. a** Probabilistic fear discrimination procedure. **b** Representative neuropixels implant. **c** Cue suppression ratios during recording sessions (n = 73). **d** Summary of brainstem regions recorded. **e** Number of single units recorded from each brainstem region by sex (n = 1812, female n = 965). + indicates 95% bootstrap confidence interval does not contain zero. d danger, u uncertainty, s safety. Source data are provided as a Source Data file.

and held 1812 neurons during 73, 1-h recording sessions (965 neurons from the 4 females, Supplementary Table S1). Neurons spanned 21 brainstem regions[13] (Fig. 1d), including subregions and neighboring regions of the superior colliculus, periaqueductal gray, dorsal raphe, and median raphe (Fig. 1e and Supplementary Table S2).

Brainstem neurons showed marked cue firing that varied in time course, direction, and pattern (Fig. 2a). K-means clustering for mean, single-unit firing in 1-s bins from 2-s prior to 2-s following cue presentation (danger, uncertainty, and safety) revealed neurons could be organized into at least 21 functional clusters; potential building blocks for brainstem construction of threat probability and behavior. Cluster size varied (min size = 24, max = 219, and median = 83) and consistent firing themes emerged when clusters were visualized (Fig. 2b and Supplementary Figs. S3 and S4). Many clusters showed ordered cue firing that differentiated danger, uncertainty, and safety (e.g., k1, k2, k6, and k7). Principal component analysis (PCA) revealed ordered cue firing (danger > uncertainty > safety) to be the primary low-dimensional feature across all brainstem neurons (PC1, explaining 29.7% of firing variance; Fig. 2c, inset).

We used an iterative, PCA shuffle analysis to determine the magnitude of each cluster's PC1 contribution. Cue firing for the neurons comprising a specific cluster (e.g., k1) was shuffled, while cue firing for the neurons of all other clusters was left intact (e.g., k2–k21). Shuffling and PCA were performed 1000 times per cluster. The change in % explained firing variance from the complete data (29.7%) to the shuffled data was calculated and averaged across the 1000 iterations for each specific cluster [PC1 complete − mean (PC1 k1 shuffled)]. PC1 contribution per unit was obtained by dividing the final value by the number of neurons in the cluster. Clusters contributing more to PC1 have higher values.

PC1 firing information originated from nine clusters (k1–k9; Fig. 2c and Supplementary Fig. S5) composed of a minority of neurons (505/1812, 27.9%). Clusters k1–k9 showed shorter firing latencies following danger onset, and across all clusters, danger firing latency negatively correlated with the magnitude of PC1 contribution ($R^2 = 0.41$, $p = 0.0017$; Fig. 2d). Clusters k1–k9 further separated themselves based

on cluster-cluster cue firing correlations (Fig. 2e and Supplementary Fig. S5), forming a functional subnetwork within the larger brainstem network. An exception was k3, which showed weak firing correlations with fellow clusters. K1 and k2 neurons showed indicators of a subnetwork hub: having the greatest PC1 contributions, shorter danger firing latencies (Fig. 2d), and strong firing correlations with the majority of their fellow clusters. Furthermore, k1 and k2 single-unit firing correlated most strongly with mean cue firing of their fellow subnetwork clusters (Fig. 2f and Supplementary Fig. S5). Neurons from each cluster were observed in at least six brainstem regions (Fig. 2g). Subnetwork neurons—including k1 and k2 hub neurons—were concentrated in the deep layer of the superior colliculus and subdivisions of the periaqueductal gray.

Ordered cue firing is the predominant brainstem feature. Yet, ordered cue firing could reflect fear behavior or threat probability. Cue firing reflecting behavior should scale to the level of suppression, invariant of shock probability. Cue firing reflecting threat probability should linearly scale with shock probability (0.0, 0.25, and 1.0), invariant of the level of suppression. Linear regression revealed unique fear behavior and threat probability signaling across the 21 clusters (Fig. 3a). Clusters showed considerable temporal variation in signaling prior to and following cue presentation, characterized by greatest signaling at onset (e.g., k5), offset (e.g., k15), sustained over cue presentation (e.g., k10), or even U-shaped signaling peaking mid cue (e.g., k6).

To reveal low-dimensional signaling features across all clusters (and therefore across the brainstem), we performed PCA on fear behavior and threat probability beta coefficients prior to and following cue presentation (Fig. 3b, top). PC1 reflected a sustained behavior signal that peaked mid cue presentation, with lesser and opposing threat probability signaling (62.7% of signaling variance; Fig. 3b, middle). PC2 reflected a sustained threat probability signal that peaked during early cue presentation, with lesser behavior signaling (26.12% of signaling variance; Fig. 3b, bottom, Supplementary Fig. S5). Thus, stable signals for fear behavior and threat probability emerged from disparate, temporal signals across clusters.

To reveal network-specific contributions to low-dimensional signals, we iteratively shuffled or "lesioned" cluster firing for one network (e.g., subnetwork clusters k1–k9), while leaving the remaining clusters intact (e.g., supranetwork clusters k10–k21). After network-specific firing shuffling, we performed linear regression for each cluster, then performed PCA for behavior and threat probability beta coefficients across all clusters. Comparing intact signaling (Fig. 3b), to signaling observed with the subnetwork lesioned (Fig. 3c), versus the supranetwork lesioned (Fig. 3d), allowed us to determine the relative contributions of each brainstem network to behavior and threat probability signaling.

Sustained threat probability signaling depended more on the cue subnetwork, while sustained fear behavior signaling depended more on the cue supranetwork (Fig. 3c, d). Lesioning the subnetwork left sustained behavior signaling intact (Fig. 3c, PC1, middle), but reduced threat probability signaling, most apparent at cue onset (Fig. 3c, PC2, bottom). By contrast, lesioning the supranetwork diminished sustained behavior signaling (Fig. 3d, PC1, middle), and shifted sustained threat probability signaling to dynamic, probability-to-behavior signaling (Fig. 3d, PC2, bottom). Neurons contributing to the subnetwork and supranetwork were distributed throughout the brainstem. Subtle anatomical biases were only apparent for the cue subnetwork (Fig. 3e), with the deep layer of the superior colliculus and the lateral subdivision of the periaqueductal gray (also sources of hub neurons) contributing more to the cue subnetwork. These findings reveal the brainstem is composed of diverse, neuronal building blocks whose specific cue firing patterns (Fig. 2a) carry unique temporal information about threat probability and fear behavior (Fig. 3a). A sustained fear behavior signal is observed across all brainstem neurons (Fig. 3c,

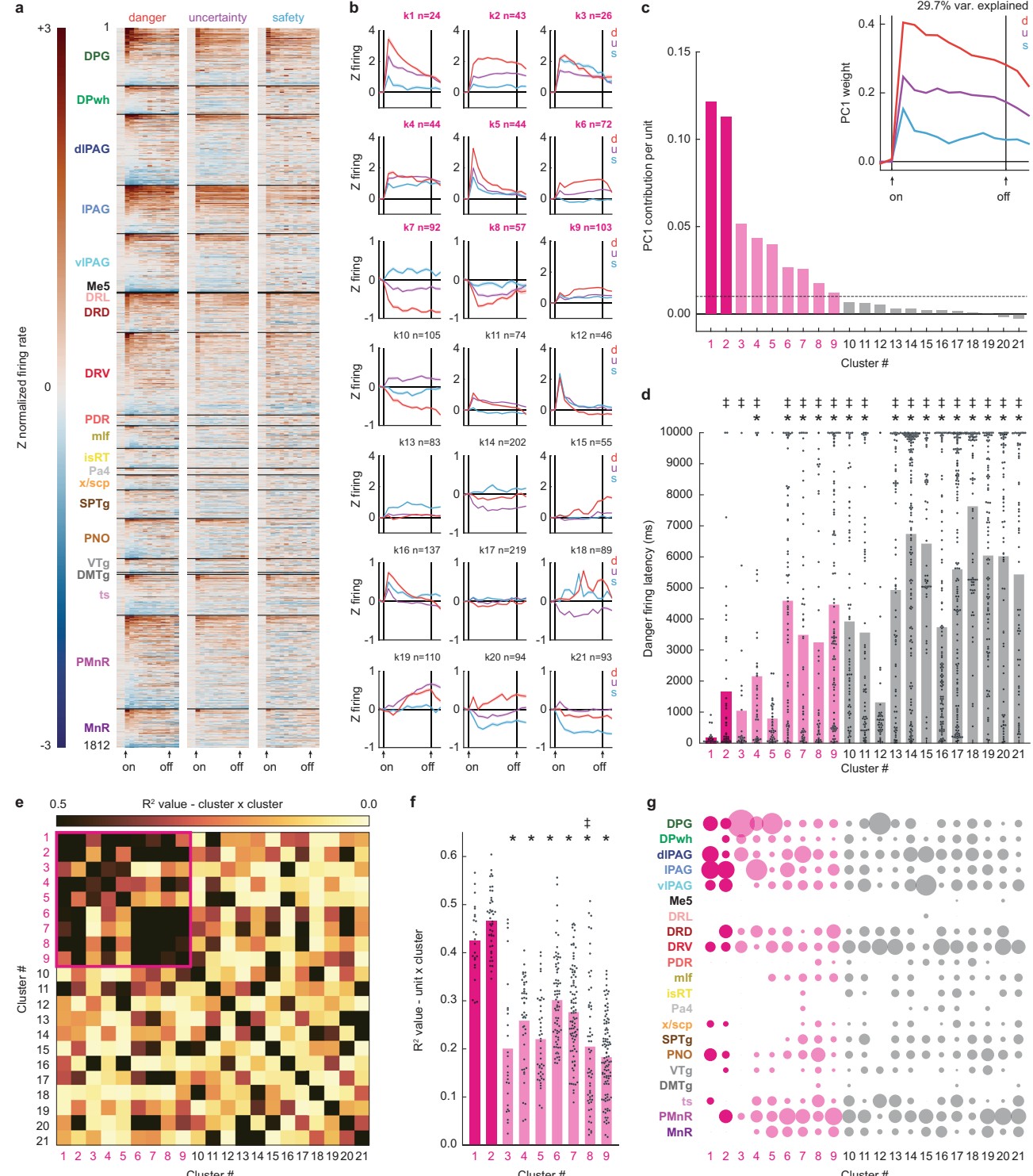

**Fig. 2 | Brainstem cue firing. a** Single-unit firing ($n = 1812$) to danger, uncertainty, and safety cues organized by brain region, dorsal to ventral. **b** Mean cluster (k1–k21) firing over cue presentation (cluster $n$ shown in the figure). Data are presented as mean values ± SEM. **c** PC1 for brainstem cue firing (inset) and cluster contribution to PC1. **d** Single-unit danger firing latency with cluster means. **e** Between-cluster cue firing correlations (data from **b**). Cue subnetwork outlined in magenta. **f** Single unit × cluster firing correlations for the cue subnetwork, with cluster means. **g** Proportion of each cluster found in each brainstem region. **c**–**g** Cue subnetwork hubs are magenta, cue subnetwork pink, and cue supranetwork gray. *Significance of two-tailed, independent samples $t$-test, Bonferroni corrected. ‡Significance of two-tailed, Levene's test for equality of variance, Bonferroni corrected. d danger, u uncertainty, s safety. Source data are provided as a Source Data file.

middle) while a sustained threat probability signal (Fig. 3d, bottom) is constructed by functional subnetwork.

Fear behavior and threat probability signals are shaped by prediction errors generated following surprising shock delivery and omission. To capture prediction error-related firing, we focused on the

10 s following shock offset. Brainstem neurons showed marked and varied firing changes, particularly following "surprising" shock on uncertainty trials (Fig. 4a). K-means clustering for mean post-shock firing (4 trial types: danger, uncertainty shock, uncertainty omission, and safety) revealed brainstem neurons could be organized into at

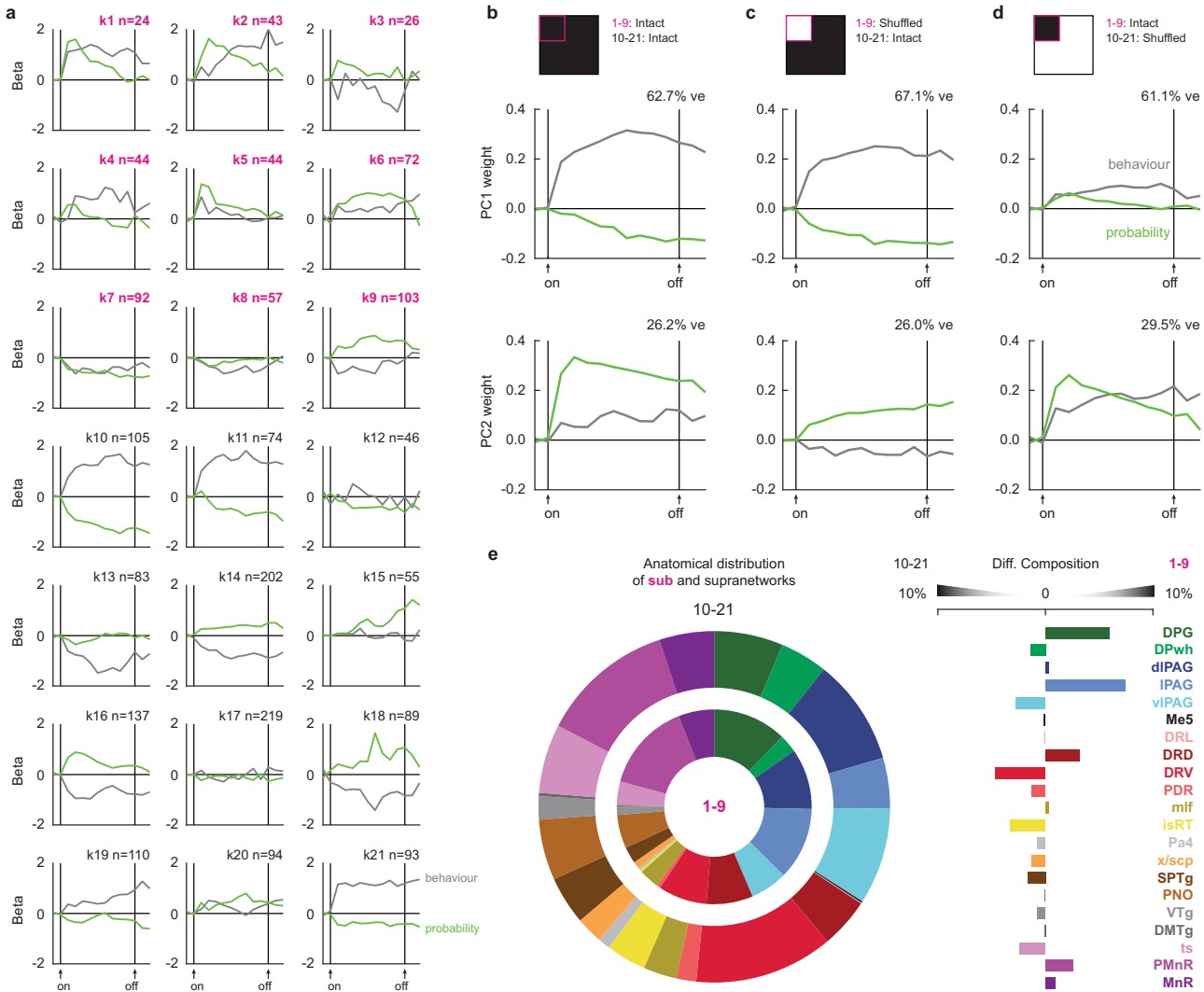

**Fig. 3 | Brainstem threat probability and behavior signaling. a** Mean cluster (k1–k21) beta weights for threat probability and behavior over cue presentation (cluster *n* shown in the figure). **b** Principal components for cluster beta weights (top; data from **a**), resulting PC1 (middle), and PC2 (bottom). Data are presented as mean values ± SEM. **c** Principal components for cluster beta weights with cue subnetwork shuffled (top; data from **a**), resulting PC1 (middle), and PC2 (bottom).

**d** Principal components for cluster beta weights with cue supranetwork shuffled (top; data from **a**), resulting PC1 (middle), and PC2 (bottom). **e** Proportion of cue subnetwork and supranetwork single units by brain region (left), and differential composition of subnetwork and supranetwork by brain region (right). ve variance explained. Source data are provided as a Source Data file.

least 11 functional clusters (min size = 46, max = 322, and median = 169; Fig. 4b and Supplementary Fig. S6). PCA for mean post-shock firing of all 1812 brainstem neurons revealed positive prediction error, on top of shock responding, to be the primary low-dimensional feature (PC1 explains 17.7% of firing variance, Fig. 4c and Supplementary Fig. S7). The prediction error is "positive" because the PC1 weight for surprising shock delivery (on uncertainty trials) exceeded the PC1 weight for predicted shock (on danger trials). The prediction error is not "signed" because it did not include an opposing, negative error—greater firing decreases to surprising shock omission (on uncertainty trials) versus predicted omission (on safety trials).

Unlike the cue period, the temporal pattern of shock firing organized outcome clusters. K1–k3 neurons showed phasic firing changes following shock. K1 neurons showed the greatest, phasic firing increases to surprising shock, and k2 neurons greatest, phasic firing increases to predicted shock, while k3 neurons physically suppressed firing following shock irrespective of trial type (Fig. 4b, left column). Cluster-cluster firing correlations revealed k1–k3 formed a phasic outcome network (Fig. 4d and Supplementary Fig. S7). K6–k9 neurons showed sustained firing changes following surprising shock (Fig. 4b,

right column). K6 neurons selectively inhibited firing to surprising shock, while k7 and k8 neurons showed sustained firing increases that were maximal to surprising shock. Cluster-cluster firing correlations revealed k6–k9 formed a tonic outcome network (Fig. 4d). Network-specific PCA for post-shock firing revealed equivalent and phasic shock firing on danger and uncertainty trials to be the primary low-dimensional feature of the phasic outcome network (explaining 37.0% of firing variance; Fig. 4e, left). PCA further revealed selective firing to surprising shock, and opposing firing to safety, to be the primary low-dimensional feature of the tonic outcome network (explaining 29.2% of firing variance; Fig. 4e, right).

Within the phasic outcome network, k1 and k2 single-unit firing was better correlated with population firing of their fellow network clusters than were k3 neurons (Fig. 4f, left). It is problematic to describe k1 and k2 neurons as hubs, given that the total network contains only 3 clusters. K6 neurons were a hub for the tonic outcome network. K6 neuron firing, the cluster decreasing firing to surprising shock, correlated most strongly with population firing of their fellow tonic outcome clusters (Fig. 4f, right). Neurons comprising the phasic and tonic outcome networks differed somewhat in their anatomical

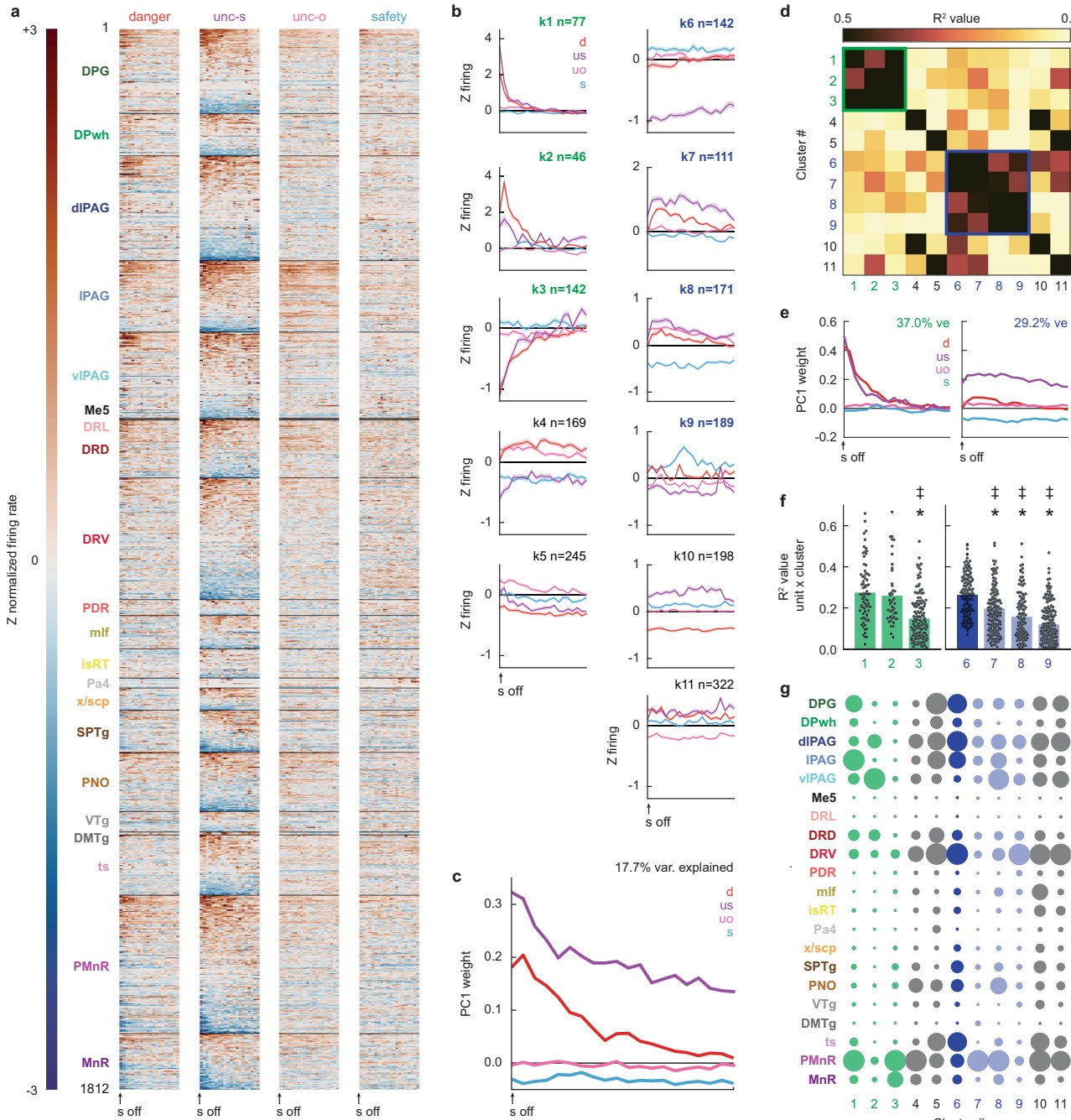

**Fig. 4 | Brainstem outcome firing. a** Single-unit firing (n-1812) following danger, uncertainty shock, uncertainty omission, and safety organized by brain region, dorsal to ventral. **b** Mean cluster (k1–k11) firing following shock (cluster *n* shown in the figure). **c** PC1 for brainstem outcome firing. **d** Between-cluster cue firing correlations (data from **b**). Phasic outcome network outlined in green, tonic outcome network outlined in blue. **e** PC1 for phasic outcome network firing (left), and tonic outcome network firing (right). **f** Single unit × cluster firing correlations for the phasic outcome network (left, green), and tonic outcome network (right, hub dark blue and network light blue). **g** Proportion of each cluster found in each brainstem region (color meaning as in **f**). *Significance of two-tailed, independent samples *t*-test, Bonferroni corrected. ‡Significance of two-tailed, Levene's test for equality of variance, Bonferroni corrected. d danger, us uncertainty shock, uo uncertainty omission, s safety. Source data are provided as a Source Data file.

distribution (Fig. 4g). Phasic outcome neurons were more common at axis extremes: subregions of the periaqueductal gray, dorsal raphe, and paramedian raphe. Tonic outcome neurons were distributed throughout the brainstem, a pattern most striking for k6 hub neurons.

We turned to linear regression in order to distinguish brainstem signals for sensory shock (equating shock firing on danger and uncertainty trials), versus prediction error (differential shock firing on uncertainty trials compared to danger). First, cluster-specific linear regression revealed unique shock and prediction error signals across

the 11 outcome clusters (Fig. 5a). For example, k1 neurons transiently signaled both sensory shock and prediction error, while k6 neurons exclusively signaled error. PCA for shock and prediction error beta coefficients across all clusters (therefore all brainstem neurons) revealed opposing sensory shock and prediction error signals to be the primary low-dimensional feature (PC1, 53.7% of signaling variance; Fig. 5b, middle). PC2 reflected dynamic, sensory shock to prediction error signaling (33.4% of signaling variance; Fig. 5b, bottom and Supplementary Fig. S7).

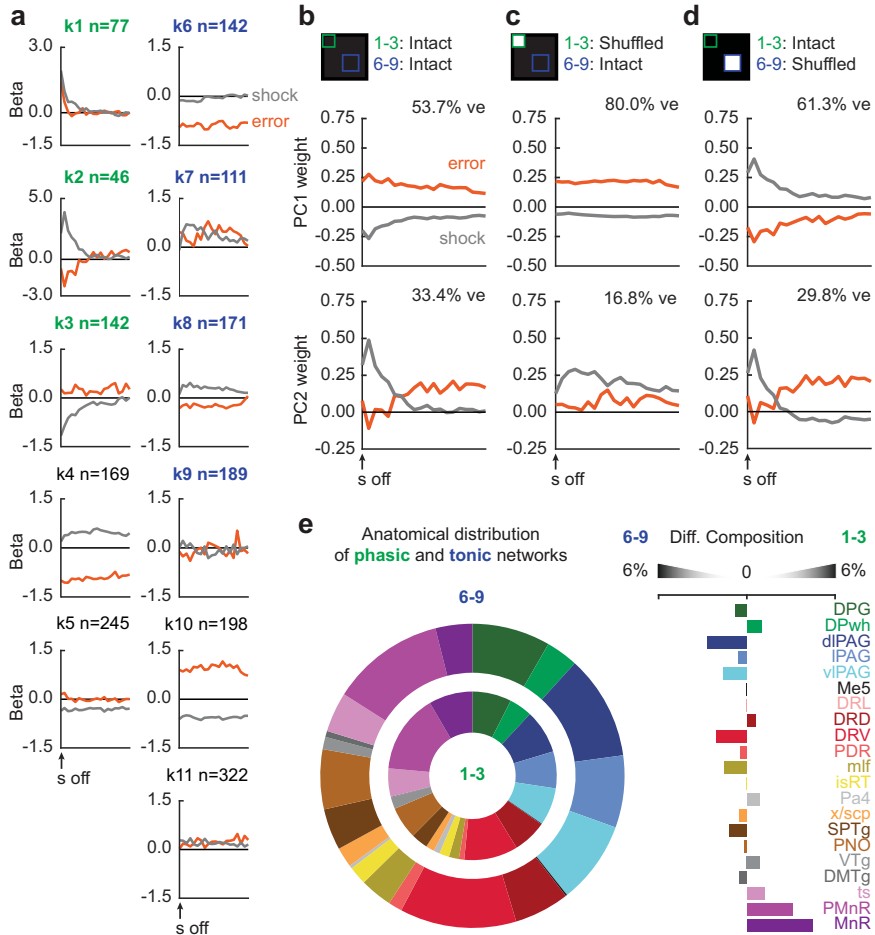

**Fig. 5 | Brainstem prediction error and shock signaling. a** Mean cluster (k1–k11) beta weights for shock and prediction error following shock delivery and omission (cluster *n* shown in the figure). **b** Principal components for cluster beta weights (top; data from **a**), resulting PC1 (middle), and PC2 (bottom). **c** Principal components for cluster beta weights with phasic outcome network shuffled (top; data from **a**), resulting PC1 (middle), and PC2 (bottom). **d** Principal components for cluster beta weights with tonic outcome network shuffled (top; data from **a**), resulting PC1 (middle), and PC2 (bottom). **e** Proportion of phasic and tonic network single units by brain region (left), and differential composition of tonic and phasic network by brain region (right). ve variance explained. Source data are provided as a Source Data file.

Rapid sensory shock signaling depended on the phasic outcome network (Fig. 5c). Lesioning the phasic outcome network (using a PCA lesion approach identical to the cue networks) left signaling dominated by sustained prediction error (80.0% of signaling variance; Fig. 5c, middle), while the residual reflected sustained shock (16.8% of signaling variance; Fig. 5c, bottom). By contrast, lesioning the tonic outcome network emphasized opposing, phasic sensory shock and prediction error signaling (61.3% of signaling variance; Fig. 5d, middle). Residual signaling reflected dynamic, sensory shock to prediction error (29.8% of signaling variance; Fig. 5d, bottom), similar to that observed across all brainstem neurons (Fig. 5b, bottom). Thus, a phasic brainstem network signals sensory shock and prediction error from a subset of neuronal building blocks transiently responsive following shock. Concurrently, a tonic brainstem network preferentially constructs positive prediction error from a subset of neuronal building blocks sustaining activity following shock. Anatomical biases for phasic and tonic outcome network neurons were absent (Fig. 5e).

The same 1812 neurons constructing threat probability and behavior during cue presentation, signaled shock and prediction error following shock delivery. We were curious whether there was a relationship between network membership during cue and outcome periods. Recall that the cue subnetwork was composed of 505 neurons (27.9% of all neurons), many fewer neurons than the cue supranetwork (1307/1812, 72.1%). Given this imbalance, the relevant question is if the

proportion of phasic ($n = 265$) and tonic ($n = 613$) outcome neurons contributing to the cue subnetwork differ from 27.9%. Chi-squared testing found neither the phasic outcome neurons (85/265, 32.1%; $\chi^2 = 2.27$, $p = 0.13$) nor the tonic outcome neurons (159/613, 25.9%; $\chi^2 = 1.17$, $p = 0.28$) differed from the expected proportion of 27.9%. A neuron's membership to cue networks constructing threat probability and fear behavior had no influence on membership to outcome networks signaling sensory shock and prediction error.

## Discussion

We set out to reveal brainstem signaling of threat probability versus fear behavior. Supporting a prevailing view[1], we observed functional populations whose firing was better captured by trial-by-trial fluctuations in behavior, rather than threat probability. The firing independence of these populations and their diffuse anatomical distribution meant continuous brainstem behavior signaling from cue onset through shock delivery. Opposing the prevailing view, functional populations whose firing was better captured by threat probability were observed. Continuous brainstem threat probability signaling was also achieved by anatomically diffuse functional populations. However, an organized threat probability signal was uncovered. Brainstem populations showing pronounced differential firing to danger, uncertainty, and safety, plus short-latency danger firing changes, formed a functional subnetwork. Subnetwork "hub" neurons were concentrated

in the deep layer of the superior colliculus and periaqueductal gray. Rather than being the exclusive domain of the forebrain, the brainstem constructs threat probability.

We further found that prediction error signaling is fundamental to the brainstem. This is consistent with prior studies which have reported prediction error in the periaqueductal gray[7,8,14]. However, our results paint a more complex picture. First, the brainstem contains two networks operating on different timescales. A phasic outcome network is engaged following shock, with composing functional populations inhibited to shock, or showing selective firing increases to surprising or predicted shock. Populations for a surprising shock—positive error—overlap with known centers for prediction error generation[8]. A tonic outcome network signals a positive prediction error. Unique to the tonic outcome network: composing functional populations—even the hub—are anatomically distributed. Even more, hub neurons show preferential firing decreases to surprising shock. The results suggest either multiple aversive prediction error systems in the brainstem, or a single system in which a tonic error signal opens a window of permissibility for phasic error to update threat estimates[15].

Construction is an apt term for brainstem threat probability and prediction error signaling. A remarkable feature of single-unit responding during the cue and outcome periods was its diversity: direction, magnitude, trial-type pattern, and temporal characteristics. Single-unit responding was diverse—not chaotic. Order first emerged in population firing: single units with similar function were observed across regions. Functional populations were not isolated, but showed specific patterns of correlated firing. The result was a consistent, network-level signal for threat probability during the cue period; and signals for prediction error during the outcome period. Stable network-level brainstem signals are constructed from disparate neuronal building blocks.

A caveat to our prediction error findings is that we did not observe population-level or brainstem-level signaling of negative error. The intermediate levels of nose poke suppression to the uncertainty cue confirm that shock omission was detected—otherwise, uncertainty behavior would have been equivalent to danger. Though because we selected a 25% foot shock probability for the uncertainty cue, shock trials were rarer than omission trials. There is evidence that midbrain dopamine neurons signaling prediction error are sensitive to rare outcomes[16]. Had we made shock omission rarer than shock delivery (e.g., using a 75% shock probability), neural correlates of negative error might have emerged. Yet given that a 50% shock probability cue can support behavior comparable to a 100% cue[12], negative error generated by surprising omission to a 75% shock cue would be insufficient to weaken cue-shock associations and reduce fear behavior.

Viewing the forebrain as the source of threat estimation has meant continuous refinement of forebrain threat processing. Cortical subregions are being linked to specific threat functions[17]. Amygdala threat microcircuits are being mapped in intricate detail[18]. Brainstem regions contain the building blocks needed to construct threat estimates. This finding necessitates refinement and detail of brainstem threat function on par with its forebrain counterparts. Expanding on prior brainstem work[19–21], our results reveal the superior colliculus[22–24] and periaqueductal gray[25–27] as prominent sources of threat information. We reveal abundant and diverse threat signaling in the paramedian raphe[28], an unstudied region adjacent to the serotonin-containing median raphe. Critically, these regions do not function in isolation. Rather, the superior colliculus and periaqueductal gray organize a local brainstem network to signal threat probability.

Perhaps the brainstem signals threat probability, but this signal is trained by the forebrain. This would be consistent with our findings. Yet, where do forebrain threat estimates originate, and once formed, how are threat estimates updated? Prediction error provides a plausible mechanism for forming and updating threat estimates. Preferential responding to surprising aversive events—consistent with positive prediction error—has been reported in many human forebrain regions[29]. Preferential responding to surprising omission of aversive events—consistent with negative error—has also been observed in human forebrain regions[30]. However, opposing firing changes to positive and negative error in the same region—a requirement of a signed prediction error[31,32]—are more narrowly observed in the human brainstem[8]. Direct manipulation of error-related activity in the brainstem of rats and mice alters fear behavior[3,7,33–35]. Available evidence suggests that learned threat estimates originating in the forebrain require prediction error generated in the brainstem. In which case, de novo acquisition of a brainstem threat estimate, trained by the forebrain, would require brainstem-generated prediction error. Also plausible—brainstem-generated prediction error may train and update a brainstem threat estimate, bypassing the forebrain altogether.

Fully revealing the brain basis of threat computation is essential to understanding adaptive and disordered fear. Our finding of widespread and organized brainstem threat signaling calls for the abandonment of the historical division of labor view. In its place, we must embrace a brain-wide view of threat computation[36–38] in which brainstem networks are not limited to organizing fear behavior but are integral to estimating threat.

## Methods

### Subjects
Subjects were six male and four female Long-Evans rats, split over two rounds of testing. The first round included three female and two male rats born in the Boston College Animal Care facility, housed with mothers until postnatal day 21 when they were weaned and single housed. The second round included four males and one female, obtained from Charles River weighing 250–275 g on arrival. All were maintained on a 12-h light-dark cycle (lights on 0600–1800) and were aged between 95 and 140 days old at the time of the first recording session. All protocols were approved by the Boston College Animal Care and Use Committee, and all experiments were carried out in accordance with the NIH guidelines regarding the care and use of rats for experimental procedures.

### Behavioral apparatus
Training took place in individual sound-attenuated enclosures that each housed a behavior chamber with aluminum front and back walls, clear acrylic sides and top, and a metal grid floor. Each grid floor bar was electrically connected to an aversive shock generator (Med Associates, St. Albans, VT) through a device that ensured the floor was always grounded apart from during shock delivery. A single food cup and central nose poke opening equipped with infrared photocells were present on one wall. Auditory stimuli were presented through two speakers mounted on the enclosure ceiling. Auditory cues were 10 s in duration and consisted of repeating motifs of a broadband click, phaser, or trumpet, which previous studies have found to be discriminable and equally salient. Testing took place in an identical chamber, but was equipped with a custom plastic food cup, plastic front and back walls, and multi-axis counterbalanced lever arm (Instech Laboratories, MCLA) with plastic tubing that held the recording cable and entered the chamber via a custom plastic top.

### Nose poke acquisition
Rats were food restricted to 85% of their free-feeding body weight, with ad-libitum access to water. After pre-exposure to pellets (Bio-Serv, Flemington, NJ) in their home cages for 2 days, rats were shaped to nose poke for pellet in the experimental chamber. During the first session, the nose poke port was removed, and rats were issued one pellet every 60 s for 30 min. In the next session, the port was reinserted, and poking was reinforced on a fixed ratio 1 schedule in which one nose poke yielded one pellet until they reached ~50 nose pokes or 30 min. Nose poking was then reinforced on a variable interval 30-s (VI-

30) schedule for one session, then a VI-60 schedule for the next four sessions. The VI-60 reinforcement schedule was utilized during subsequent fear discrimination and was independent of auditory cue and shock presentation.

## Fear discrimination

Rats received 12 sessions of Pavlovian fear discrimination prior to Neuropixels implant. Each 54-min session consisted of a 5-min warm-up period in the chamber followed by 16 cue presentation trials. Each auditory cue predicted a unique shock probability (0.5 mA, 0.5 s): danger, $p = 1.00$; uncertainty, $p = 0.25$; and safety, $p = 0.00$. Shock was administered 2 s following the termination of the cue on danger and uncertainty-shock trials. A single session consisted of 4 danger, 2 uncertainty-shock, 6 uncertainty-no shock, and 4 safety trials with a mean inter-trial interval of 3 min. Trial order was randomly determined by the behavioral program and differed for each rat, in every session. The physical identities of the auditory cues were counterbalanced across individuals. Following recovery from surgery, rats received one VI-60 session to habituate to being connected to the recording cable. Rats then received between 1 and 10 discrimination sessions during which single-unit activity was recorded.

## Calculating suppression ratios

Timestamps for cue presentations, shock delivery, and nose pokes (photobeam break) were automatically recorded by the Med Associates program. Baseline nose poke rate was calculated for each trial by counting the number of pokes during the 20-s pre-cue period and multiplying by 3. Cue nose poke rate was calculated for each trial by counting the number of pokes during the 10-s cue period and multiplying by 6. Nose poke suppression was calculated as a ratio: (baseline poke rate − cue poke rate) / (baseline poke rate + cue poke rate). A suppression ratio of "1" indicated complete suppression of nose poking during cue presentation relative to baseline. A suppression ratio of indicated "0" indicates equivalent nose poke rates during baseline and cue presentation. Gradations in suppression ratio between 1 and 0 indicated intermediate levels of nose poke suppression during cue presentation relative to baseline.

## Surgery

Following the 12th discrimination session, rats were returned to ad-libitum food access and underwent stereotaxic surgery performed under isoflurane anesthesia (1–5% in oxygen). Four screws were screwed into the skull around the target cap area to aid adhesion of the cap, and the skull was also scored in a crosshatch pattern. A craniotomy with a 1.4 mm diameter was carried out, and the underlying dura fully removed to expose the cortex. Immediately prior to implant the probe was painted with DiI to later identify histology tracks (ThermoFisher, V22886). To maximize recording regions, each implant was aimed at coordinates −8.00 AP, −2.80 ML, −7 to −7.5 DV, with a 15° angle. Each Neuropixels probe (1.0 probe) and head stage were secured in a pre-prepared custom head cap. The cap was held and slowly lowered during implant using a modified stereotaxic arm until the max DV was reached, or until the cap contacted the skull. The craniotomy was sealed using silicone gel (Dow DOWSIL 3-4680). Once the cap was in place, the ground wire was wrapped around the two screws positioned laterally to the cap to ground the probe. Vacuum sealing grease (Dow Corning) was applied around the base of the cap to fill any space between the cap and the skull and protect the probe. Caps were cemented into place using orthodontic resin (cc 22-05-98, Pearson Dental Supply), and the head cap lid was secured in place on the head cap. Rats were given one week to recover with prophylactic antibiotic treatment (cephalexin, Henry Schein Medical) prior to data acquisition and received carprofen (5 mg/kg) for postoperative analgesia.

## Data acquisition

Neural data were recorded using Open Ephys with the Neuropixels PXI plugin running on an acquisition computer connected to the PXI chassis (PXIe-1071) containing the Neuropixels base station. Behavior events were controlled and recorded by a separate computer running Med Associates software. To get behavior timestamps, signals were sent from Med Associates to the NIDAQmx Open Ephys plugin, via Med Associates TTL adapter boxes (SG-231) plugged into a connector block (National Instruments, BNC 2110) connected to an I/O module (PXI-6363) in the PXI chassis. During recording sessions, the cable was first connected to the head stage and the head stage lid fixed in place, then the recording channels and reference for that session and subject were selected. To maximize the acquisition of neurons from the midbrain region, the channels selected were either the lowest bank of 384 channels, or channels 193–575, used in a double alternating order across sessions and counterbalanced across subjects. The external reference was selected unless that proved ineffective in which case the tip reference of the probe was used instead. After this, the doors to the chamber were closed and the fear discrimination and recording session started. Sessions were only included for analysis if the probe signal was maintained throughout all 16 trials, if the signal was lost for any reason that session was discarded. Subjects were recorded from daily up to either ten total recording sessions, or until data was no longer able to be acquired from a subject.

## Probe retrieval

Following recording sessions, rats were placed back into the stereotaxic frame under isoflurane anesthesia. The head cap lid was removed, the ground wire cut, and the head stage disconnected and removed. The cement securing the probe holder in place was scraped away with a scalpel blade and the holder slowly pulled up and out of the cap. The probe was then rinsed and soaked in DI water, followed by a soak in a tergazyme solution before a final rinse with DI water before and if still functional after explant safely stored for re-implant.

## Histology

Once the probe had been explanted, the rat was removed from the frame and deeply anesthetized using isoflurane before being perfused intracardially with 0.9% biological saline and 4% paraformaldehyde in a 0.2 M potassium phosphate buffered solution. Brains were extracted and fixed in a 10% formalin solution for 24 h, then stored in 10% sucrose/formalin. Brains were sliced with a microtome into 40 μm sections (from approximately Bregma −6.5 to −9, to ensure the full extent of the probe tracks could be identified). The tissue was rinsed, incubated in NeuoroTrace (ThermoFisher, N21479), rinsed again, and then mounted prior to imaging within a week of processing (Axio Imager, Z2, Zeiss) to locate probe placement using the visible DiI tracks and NeuroTrace. Neuron locations were established by identifying the 3D location of the tip of the probe relative to the Allen Atlas, as well as the location in which the probe entered the brain (not including the cortex) and the vector of implant calculated. These were also checked against expected electrophysiology patterns (regions of expected low and high activity) for location accuracy.

## Neuron sorting

Data were automatically spike sorted using Kilosort 2[39] or 2.5 (https://github.com/MouseLand/Kilosort). Clusters identified by Kilosort were manually curated in Phy (https://github.com/cortex-lab/phy). To assess if activity reflected single-unit activity, inter-spike interval, waveform shape, firing rates, activity change across channels, were all examined. Neurons were also assessed for potential merges with similar nearby clusters, and for potential splitting out of noise/other neurons. Neurons were only kept for analysis if the pattern of activity was confidently identified as neuron activity, and not noise or multi-unit activity. Accepted neurons were finally screened using Matlab and

only kept if consistently recorded throughout all trials in the session recorded in, any neurons that showed clear drop-offs or loss of recordings were discarded. A total of 52% of neurons accepted in Phy passed Matlab screening. Complete descriptions of spike sorting are provided in the Supplementary information.

## Analysis

Matlab was used to extract, collate, and analyze the single-unit data and behavior timestamp events. Fear was measured by suppression of rewarded nose poking (baseline poke rate – cue poke rate)/(baseline poke rate + cue poke rate). Perceptually uniform color maps were used to prevent visual distortion of the data[40]. K-means clustering was performed by systematically varying the number of clusters and examining the output for over/under clustering. Single-unit and population firing analyses utilized k-means clustering, PCA, linear regression combined with iterative shuffling. Complete descriptions of single-unit analyses are provided in the Supplementary information.

## Reporting summary

Further information on research design is available in the Nature Research Reporting Summary linked to this article.

## Data availability

The single-unit data generated in this study have been deposited at https://crcns.org/. Source Data are provided with this paper.

## Code availability

Code for single-unit analyses is available at https://crcns.org/.

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

## Acknowledgements

We thank Bret Judson and the Boston College Imaging Core for infrastructure/support, Dr Matthew Gardner and Dr Geoffrey Schoenbaum for advice and initial designs for 3D head cap printing, Joe Austen for help building and ordering acquisition computers, Richard Pijar and the Boston College Machine Shop for electrical and hardware support and instrument customization, Pavel Kulik and Open Ephys for assistance with the I/O module, Dr Nicholas Steinmetz, Dr Matteo Carandini and the UCL Neuropixels course for initial training in the use of silicone probes. Research reported in this publication was supported by the National Institute of Mental Health of the National Institutes of Health under Award Number R01MH117791 to M.A.M. The content is solely the responsibility of the authors and does not necessarily represent the official views of the National Institutes of Health. This work was also supported in part by an Ignite Grant from Boston College.

## Author contributions

Conceptualization: J.A.S., M.A.M. Methodology: J.A.S., M.A.M. Investigation: J.A.S., M.A.M. Funding acquisition: M.A.M. Writing—original draft: J.A.S., M.A.M. Writing—review and editing: J.A.S., M.A.M.

## Competing interests

The authors declare no competing interests.
