## [Peer Review File · Nature Communications]

Brainstem networks construct threat probability and prediction error from neuronal building blocksREVIEWER COMMENTS

Reviewer #1 (Remarks to the Author):

This manuscript examines single unit spiking activity in the rat brainstem in order to address the question of whether and how the brainstem may be involved in threat assessment and prediction. Briefly, traditional interpretations of how the brain responds to threats have suggested that the forebrain and neocortex are primarily responsible for assessing threat probability and prediction error, while the brainstem is involved in the associated behavior. This view has been challenged by recent literature providing evidence that indeed the brainstem is also involved in assessing threat probability and prediction error. Here, the authors use Neuropixels probes to capture spiking activity from a large population of neurons in the rat brainstem as the rats engage in a probabilistic fear discrimination task. Their results demonstrate that different populations of neurons, or subnetworks, are involved in assessing threat probability, modulating behavior, and registering outcomes and prediction errors. The results therefore provide additional evidence that threat assessment may be a brain-wide phenomenon, rather than one that is restricted to individual brain regions like the forebrain. The data presented and the results provide strong evidence for this conclusion, as fundamentally they show that individual neurons and clusters of neurons exhibit clear modulation in their activity that is related to threat uncertainty and outcome. As such, it appears that their conclusions are reasonably supported, although there are some suggestions that may improve the overall strength of the conclusions.

One issue that could be addressed is how clusters of neurons that comprise the various subnetworks are constructed, particularly across animals. The authors effectively pool all of the spiking responses from all neurons together before performing a clustering analysis. Based on this approach, they find clusters that differentially respond to threat uncertainty and behavior and to outcome and prediction error. However, it is not clear if such subnetworks or differential activity are present within individual animals, or whether this may simply reflect the possibility that different animals exhibit differential neural activity for threat probability or behavior, for example. In other words, is it possible that the subnetworks identified in figure 2 or figure 4, for example, are simply reflecting a clustering between different animals. Does k1, for example, all come from one animal? Do these same clusters exist within individuals?

The authors suggest that their data demonstrate that there is a cue subnetwork that exhibits dynamic probability to behavior signaling, while there is a cue supra-network that exhibits sustained behavioral signaling. To show this, they regress the activity of each network with either probability or behavior, and then perform a PCA analysis on the regression weights. The conclusion regarding the dynamic changes are largely based on the changes observed in the PC2 weight. However, the regression weights themselves would suggest that this distinction is not so clear (Fig 3a). There are several clusters within the subnetwork that exhibit a sustained and clear response to either probability or behavior without evidence of this dynamic. Similarly, there are several clusters in the supranetwork that exhibit preferential signaling of probability.

Similarly, the anatomic distinction between the subnetwork and supranetwork also appears less clear than the authors would suggest. This is presented in figure 3E, but here there appears to be a fair amount of overlap between anatomic regions between the two networks.

In the second half of the manuscript, the authors then focus on the neural responses to the shock, and whether neurons encode shock outcome versus prediction error. This is a nice demonstration that indeed brainstem activity is involved in prediction error. A key question, however, is whether the same clusters involved in threat probability and behavior are also involved in outcome and prediction error. The authors identify clusters independently for outcome and prediction error (the same criticism regarding individual animals versus the pooled group of neurons applies here) and then ask how much overlap there is between neurons in one subnetwork versus another using a Chi squared test. Based on this, they conclude that the cue subnetwork is distinct from the tonic outcome network. However, it appears that there are more neurons in the cue network that are involved in the tonic

outcome network (190) compared to those involved in the phasic outcome network (169). It is true that it is less likely that tonic outcome neurons are involved in the cue subnetwork, but looking at the relationship in the opposite direction would lead to the opposite conclusion. Perhaps there may be a clearer way to demonstrate the relative contributions of each subnetwork and cluster to both probability/behavior and outcome/prediction.

An interesting result is the differential response to unexpected shock, expected shock, and then omission. The authors identify the omissions during the uncertain trials as unexpected. However, given the shock schedule used in the task, one would think that these omissions should actually be classified as expected, similar to the safety trials, since they happen much more frequently than the shocks.

Finally, some of the anatomic conclusions are determined based on their method for localization. This involves locating the tip of the probe and the entry in 3D space, and then using the Allen atlas to then estimate the locations of each electrode. Since the authors have histology, is it possible to identify the actual location of each probe, and therefore each electrode, directly from the individual rat brainstem based on the histological slices?

Reviewer #2 (Remarks to the Author):

This is an interesting study describing the role of brainstem in construction of threat probability, behaviour, and prediction error. To this end the authors used neuropixels probes to record single-unit activity across a 21-region brainstem axis during probabilistic fear discrimination. They identified a dorsally-based brainstem network rapidly signaled threat probability.

The article is clear and well written, but the work does not completely support the conclusions and claims. The experiments and analysis seem carefully performed; yet the methodology is not always detailed and there are some questions which need to be addressed:

1) In the experimental setup and anatomical location of the Neuropixels (Fig 1) could the authors add a figure that describes the suppression ratio that is used as fear behaviour measure throughout the paper? It remains very abstract. Also, I think using the suppression of reward seeking behaviour (the rats searching for food) is suboptimal to measure fear, because it implies reward circuits. Please comment why freezing is not used for fear.

2) The values that were clustered (e.g. waveform?,...) presented in Fig 2B are not clear; could you please add precisions in the text and Figure 2?

How the number of clusters was found (was it a number that was chosen before, and if so on which criteria?) was also not clear; please add this information to better understand the number of 21 clusters.

On which trials the PCA was made (are all trials included in one PCA, and if so, how were the curves for the single experimental conditions computed?). Please add precision in the text or methods.

In these clusters, some of the smaller clusters look quite similar in their firing z-score. For example, cluster 1+5, cluster 2+8 or cluster 6+7. There might have been over clustering of the data? Please explain and discuss how baseline changes are likely to change the outcome of the z-score.

3) Data (clusters) presented in the Fig 2C are interpreted as reflecting different threat probability. However, none of the clusters shows an important difference in firing towards the uncertain or the safe stimulus. It could be that the rats simply did not understand the association of the 'uncertain' tone to the shock? Please discuss.

4) Clusters (about firing latency to danger) presented in Fig 2D seem to contain two groups of cells with different latencies. Maybe these clusters contain different subclusters? Please discuss.

5) Figure 2E show cue firing correlation between clusters. Clusters 1-8 are shown to have very high correlation. But if there was overclustering in these clusters, the correlation would only indicate that some clusters belong together. How did you checked for overclustering?

6) The clustering (Fig 2G) might be explained by the firing onset corresponding to the location of the most cells in the descending pathways from cranial to caudal. Indeed, the danger firing latency in 2D might be dependent on the location of the cells in the brainstem. In figure 2G the clusters that contained more cranial neurons had shorter firing latencies (e.g., cluster 1, cluster 10). Please discuss.

7) The authors tried to explain the firing of the single clusters over time in Fig 3 by doing a linear model, in which the threat probability and the behaviour were used as regressors. Please add methodology details for this model. Do the authors validate if the different firing rates in the different threat probability conditions are not representing the reaction to the different tones that were played in the different conditions?

8) Some of the clusters in Fig 3A seem to be correlating with both threat probability and behaviour regressors. Even if the curves representing the regressors are not identical, they seem to be mirrored around the x axis in some clusters (k6,12,19,20). Why you state that the firing that is explained by behaviour should be not affected by the threat probability?

9) In Figure 3D, the larger network is disabled and the small one still intact. The beta for the cue firing is smaller in the PC1, which means that the beta for the cue firing is also dependent on the larger network. In the text, the authors state that cue firing would be entirely dependent on the small network. Why do you think then that it was 'entirely' dependent?

10) Brainstem location of the neurons in the larger network containing cluster 9 to 21 and of the smaller cluster is presented in Figure 3E. It looks like the IPAG is more present in the smaller network. In the text, the authors state this was a proof that there is a more dorsal network that computes fast threat probability and a more ventral network that computes behavioural reaction and slow threat probability. Could you be more precise? This statement is too large. First, it is known that there is behaviour computed in both ventrolateral and dorsal PAG. Second, the higher contribution of IPAG neurons in the smaller network could also be due to the location of the neuropixel probe in the brainstem. In Figure 1A it looks as if the Neuropixel was implanted more lateral, touching the lateral PAG directly, but not the ventral part of the brainstem. Therefore, the neurons recorded from the IPAG were closer to the probe and maybe easier to identify in the data than the ones in the ventral areas. Therefore, they might be overrepresented in the network. Thus, is it valid to draw a conclusion of the location of a network on this?

11) In Figure 4 is presented the cluster firing regarding different conditions: Predicted shock, unpredicted shock, unpredicted omission, and predicted omission. The unpredicted omission occurs in the trial condition where the rat never experiences a shock following the tone. Therefore, the rat did not associate the tone to a shock in first place. If there is no expectation of a shock, how can the shock be omitted? Could it be that the reactions is to the 'safety' condition rather as a pure reaction to the tone as a negative prediction error? Please precise from which trial the 'unpredicted foot shock' originates. If the unpredicted foot shock is the foot shock from the uncertainty condition, it is not a completely unpredicted foot shock. Therefore, the positive predication error signal might be not pure in this condition. Please discuss. In Figure 4B the duration was long, 10s (after the offset). Did the authors check if the firing of the clusters could also be related to motion?

12) Linear regression model to explain the weights of either the shock or a prediction error in the clusters firing is presented in Figure 5. Could the authors add a regressor estimating the impact of motion? (to compare with the reaction to the shock?)

13) It is not clear how the authors reached the conclusion, that the brainstem is constructing threat

probability. Indeed, in the figure 1, there seems to be no significant firing difference to uncertain foot shock firing and safety firing. Could it be that the rats did not understand the association between the insecure foot shock and the tone? Even if the signalling would represent threat probability signalling, it is not proof for the construction of the signalling in the brainstem (the signals could be simply conducted by the brainstem)

Introduction to the revised manuscript

We are grateful for the thorough and thoughtful critiques from both reviewers. As we revised the manuscript, we meticulously checked our code to verify analyses. During this process we uncovered a simple, yet critical error in our trial averaging. We apologize in advance for this very detailed error description, but feel this is owed and necessary.

Our discrimination sessions consist of 16 trials (4 danger, 2 uncertainty shock, 6 uncertainty omission, and 4 safety). Trial type order is randomized so that every session is unique. To analyze firing and behavior across all sessions, we must first standardize the trial order.

Session trial order			Standardized analysis order			
Session trial #	Trial type	Trial type #	Session trial #	Trial type	Trial type #	Analysis row #
1	danger	1	1	danger	1	1
2	unc omission	1	4	danger	2	2
3	safety	1	15	danger	3	3
4	danger	2	16	danger	4	4
5	unc omission	2	7	unc shock	1	5
6	unc omission	3	13	unc shock	2	6
7	unc shock	1	2	unc omission	1	7
8	safety	2	5	unc omission	2	8
9	unc omission	4	6	unc omission	3	9
10	unc omission	5	9	unc omission	4	10
11	safety	3	10	unc omission	5	11
12	safety	4	14	unc omission	6	12
13	unc shock	2	3	safety	1	13
14	unc omission	6	8	safety	2	14
15	danger	3	11	safety	3	15
16	danger	4	12	safety	4	16

Above is an example of how this works. The three columns on the left show a hypothetical trial order for one recording session. This is the specific order that rat experienced for that session, and due to random trial selection, no other rat will receive this specific trial order. For each of the 16 session trials we determine the type (of 4 possible) and the number of occurrences for that type. We then rearrange the trial types in a standardized order for analysis. Danger trials are first, uncertainty shock second, uncertainty omission third, and safety last. Trial number is maintained within each trial type. Danger trial 1 is first, trial 2 second, etc. We perform this standardization for every session, which results in a 3D matrix: trial types are rows (X), time bins are columns (Y), and sessions are stacked layers (Z).

Many analyses require us to average all trials of a single type. This is where the error occurred. To find mean danger firing, we average rows 1:4. For uncertainty shock, rows 5:6; uncertainty omission, rows 7:12; and for safety, rows 13:16. However, when we checked our code a '1' was omitted from safety averaging. In error, we averaged rows 3:16 instead of 13:16. What we thought was mean safety firing was actually mean firing for ALL cues, except the first two danger cues. This error profoundly affected reported safety firing.

Having identified the error, we reran all analyses. The correction made obvious impacts on our results. All manuscript figures and supplements have been completely remade to reflect the accurate safety firing data. Correcting safety firing clarified and solidified our findings. Here are some of the biggest changes:

- Brainstem single units showed markedly less firing to safety than either uncertainty or danger during cue presentation (Fig. 2A).
- Functional populations showed discriminative firing that clearly distinguished danger from uncertainty and uncertainty from safety (Fig. 2B).
- Principal components analysis revealed differential cue firing to be the primary low dimensional feature, with PC weights for uncertainty falling almost exactly between danger and safety (Fig. 2C).
- Behaviour (Fig. 3B, PC1) and threat probability (Fig. 3B, PC2) signaling were now more separate low-dimensional firing features with clearer relationships to the supra and subnetworks. The cue subnetwork contributed more greatly to threat probability signaling (Fig. 3C) while the cue supranetwork contributed more greatly to fear behaviour signaling (Fig. 3D).
- Brainstem single units showed almost no firing changes to safety during the outcome period (Fig. 4A).
- Functional populations showed little safety firing during the outcome period, and those that did fire did so in a manner opposing that for uncertainty shock (Fig. 4B).
- Principal components found minimal and negative PC weights for safety (Fig. 4C).

Given this relatively simple error, we were concerned about additional errors. Particularly because many of our analyses are somewhat complex: principal components and regression combined with cluster-specific shuffling. We triple checked all code and found no additional errors. Looking back, it seems that we were so concerned with nailing down our more complex analyses, that we overlooked one of the simplest errors that could be made.

We apologize for this error. If a more detailed explanation is needed and/or if the reviewers would like to be walked through portions of the analyses – we are more than happy to comply.

Reviewer #1

This manuscript examines single unit spiking activity in the rat brainstem in order to address the question of whether and how the brainstem may be involved in threat assessment and prediction. Briefly, traditional interpretations of how the brain responds to threats have suggested that the forebrain and neocortex are primarily responsible for assessing threat probability and prediction error, while the brainstem is involved in the associated behavior. This view has been challenged by recent literature providing evidence that indeed the brainstem is also involved in assessing threat probability and prediction error. Here, the authors use Neuropixels probes to capture spiking activity from a large population of neurons in the rat brainstem as the rats engage in a probabilistic fear discrimination task. Their results

demonstrate that different populations of neurons, or subnetworks, are involved in assessing threat probability, modulating behavior, and registering outcomes and prediction errors. The results therefore provide additional evidence that threat assessment may be a brain-wide phenomenon, rather than one that is restricted to individual brain regions like the forebrain. The data presented and the results provide strong evidence for this conclusion, as fundamentally they show that individual neurons and clusters of neurons exhibit clear modulation in their activity that is related to threat uncertainty and outcome. As such, it appears that their conclusions are reasonably supported, although there are some suggestions that may improve the overall strength of the conclusions.

We appreciate your time and feedback. We address each suggestion below and in the revised manuscript.

One issue that could be addressed is how clusters of neurons that comprise the various subnetworks are constructed, particularly across animals. The authors effectively pool all of the spiking responses from all neurons together before performing a clustering analysis. Based on this approach, they find clusters that differentially respond to threat uncertainty and behavior and to outcome and prediction error. However, it is not clear if such subnetworks or differential activity are present within individual animals, or whether this may simply reflect the possibility that different animals exhibit differential neural activity for threat probability or behavior, for example. In other words, is it possible that the subnetworks identified in figure 2 or figure 4, for example, are simply reflecting a clustering between different animals. Does k1, for example, all come from one animal? Do these same clusters exist within individuals?

To address this, we have made a supplemental figure (Fig S4 & below) comparing individual identity to cluster identity across all 1,812 neurons. For the most part, cluster neurons were distributed across rats. The one exception was cluster 3, which appeared to largely come from Rat #9. Fewer units came from subjects 4, 5 and 6, the rats that completed very few recording sessions (Fig S1).

The authors suggest that their data demonstrate that there is a cue subnetwork that exhibits dynamic probability to behavior signaling, while there is a cue supra-network that exhibits

sustained behavioral signaling. To show this, they regress the activity of each network with either probability or behavior, and then perform a PCA analysis on the regression weights. The conclusion regarding the dynamic changes are largely based on the changes observed in the PC2 weight. However, the regression weights themselves would suggest that this distinction is not so clear (Fig 3a). There are several clusters within the subnetwork that exhibit a sustained and clear response to either probability or behavior without evidence of this dynamic. Similarly, there are several clusters in the supranetwork that exhibit preferential signaling of probability.

We agree it is more accurate to say that clusters that are stable and dynamic in terms of firing give rise to information that is dynamic. In the revised manuscript we explicitly point out the variety of signaling patterns – both for their temporal properties and type of information signaled [Lines 91-94]. We end the cue regression section by directly pointing out that clusters with varying properties give rise to consistent signals for behaviour and threat probability [Lines 121-125]. We also feel this better supports our ‘construction’ framing in the manuscript. These brainstem-wide signals we observe are not the product of a single functional neuron type working identically across many regions. Instead, many disparate functional neuron types signaling unique aspects of threat and behavior combine to construct a signal greater than themselves.

Similarly, the anatomic distinction between the subnetwork and supranetwork also appears less clear than the authors would suggest. This is presented in figure 3E, but here there appears to be a fair amount of overlap between anatomic regions between the two networks. In the second half of the manuscript, the authors then focus on the neural responses to the shock, and whether neurons encode shock outcome versus prediction error. This is a nice demonstration that indeed brainstem activity is involved in prediction error. A key question, however, is whether the same clusters involved in threat probability and behavior are also involved in outcome and prediction error. The authors identify clusters independently for outcome and prediction error (the same criticism regarding individual animals versus the pooled group of neurons applies here) and then ask how much overlap there is between neurons in one subnetwork versus another using a Chi squared test. Based on this, they conclude that the cue subnetwork is distinct from the tonic outcome network. However, it appears that there are more neurons in the cue network that are involved in the tonic outcome network (190) compared to those involved in the phasic outcome network (169). It is true that it is less likely that tonic outcome neurons are involved in the cue subnetwork, but looking at the relationship in the opposite direction would lead to the opposite conclusion. Perhaps there may be a clearer way to demonstrate the relative contributions of each subnetwork and cluster to both probability/behavior and outcome/prediction.

We also found it challenging to describe the relationship between a neuron’s memberships in the cue vs. outcome networks. The biggest challenge is there are many fewer cue subnetwork neurons (505/1812) than supranetwork neurons (1307/1812). Because of this, there *must* be more phasic and tonic outcome neurons in the supranetwork. The question then is if the observed proportions of each outcome network in the cue subnetwork differed from the expected proportion of 27.9%. Only now, once we correct the safety trial averaging issue and performed chi-square testing, neither proportion differed from the expected. Even though this is a null result, we believe it is an important one. Therefore, we have dedicated more space in the results section to

explaining the chi-square approach and the results [Lines 190-200].

An interesting result is the differential response to unexpected shock, expected shock, and then omission. The authors identify the omissions during the uncertain trials as unexpected. However, given the shock schedule used in the task, one would think that these omissions should actually be classified as expected, similar to the safety trials, since they happen much more frequently than the shocks.

This is valid point. With foot shock being the rarer of the two outcomes following the uncertainty cue, it might be more ‘surprising’. Further, there is evidence that midbrain dopamine – which unambiguously signal signed reward prediction errors – are additionally sensitive to rare outcomes (Rothenhoefer et al., 2021).

We see the larger point of your observation, though. Having the uncertainty cue followed by shock on 50% of trials would have made for a more balanced experimental design. In fact, the first time we tried this discrimination procedure that is exactly what we did (Berg et al. 2014 European Journal of Neuroscience). Unexpectedly, we found that a cue predicting foot shock on 50% of trials supported levels of nose poke suppression that were nearly equal to those for a 100% cue. Evolutionarily this makes sense. The goal of defensive systems is to serve, not be precisely afraid. However, in our experiments we want to see complete discrimination of our three cues. To practically achieve this, we needed to reduce the foot shock probability of the uncertainty cue to 25%. We have included this rationale in the revised manuscript [Lines 42-43].

Finally, some of the anatomic conclusions are determined based on their method for localization. This involves locating the tip of the probe and the entry in 3D space, and then using the Allen atlas to then estimate the locations of each electrode. Since the authors have histology, is it possible to identify the actual location of each probe, and therefore each electrode, directly from the individual rat brainstem based on the histological slices?

Yes, it is possible to identify the actual location of each probe. Your description of the approach is largely correct. We use the Paxinos and Watson atlas as our guide and the probe location for each subject is independently determined. For each subject, we determine the XYZ (medial-lateral, anterior-posterior, and dorsoventral) location of the tip of the probe. We then determine the XYZ location of the top of the probe (where it enters the superior colliculus typically). We then superimpose the full 3-dimensional path onto layered images from the Paxinos and Watson atlas. The detailed paths for each individual are provided in Supplemental Figure S2. The detailed path for each individual also allows us to some sanity checks. For example, single units should be sparser in regions dense with fibers of passage. We observe this pattern in single rats and across the complete data set.

Reviewer #2

This is an interesting study describing the role of brainstem in construction of threat probability, behaviour, and prediction error. To this end the authors used neuropixels probes to record single-unit activity across a 21-region brainstem axis during probabilistic fear discrimination. They identified a dorsally-based brainstem network rapidly signaled threat probability. The article is clear and well written, but the work does not completely support the conclusions and claims. The experiments and analysis seem carefully performed; yet the methodology is not always detailed and there are some questions which need to be addressed:

We appreciate your thoughtful feedback. We address each question in turn and note manuscript revisions.

1) In the experimental setup and anatomical location of the Neuropixels (Fig 1) could the authors add a figure that describes the suppression ratio that is used as fear behaviour measure throughout the paper? It remains very abstract. Also, I think using the suppression of reward seeking behaviour (the rats searching for food) is suboptimal to measure fear, because it implies reward circuits. Please comment why freezing is not used for fear.

We apologize for omitting information about suppression ratio. The revised manuscript includes more information about the use of suppression ratios in the Results section [Lines 46-49] and complete information in Methods [Lines 459-469]. Briefly, suppression ratios are calculated from baseline and cue nose poke rates: $(\text{baseline poke rate} - \text{cue poke rate}) / (\text{baseline poke rate} + \text{cue poke rate})$. A suppression ratio of '1' indicates complete suppression of nose poking during cue presentation relative to baseline. A suppression ratio of '0' indicates equivalent nose poke rates during baseline and cue presentation. Gradations between 1 and 0 indicate intermediate levels of nose poke suppression during cue presentation relative to baseline.

2) The values that were clustered (e.g. waveform?,..) presented in Fig 2B are not clear; could you please add precisions in the text and Figure 2?

We have updated the manuscript to specifically state the values used for clustering: “K-means clustering for mean, single-unit firing in 1-s bins from 2-s prior to 2-s following cue presentation (danger, uncertainty, and safety)” [Lines 57-58].

How the number of clusters was found (was it a number that was chosen before, and if so on which criteria?) was also not clear; please add this information to better understand the number of 21 clusters.

We identified '21' clusters by systematically increasing cluster number from 1 to 30. Cluster number was optimized to produce the fewest number of clusters and the smallest mean Euclidean distance of each cluster member from its centroid. For example, 1 cluster would produce the fewest number of clusters – all units belong to a single cluster. But the mean Euclidean distance of each cluster from the centroid would be very, very large. 30 clusters would mean that each unit would have a very, very short Euclidean from its centroid, but that we were dividing like neurons into separate types. A sign of overclustering is when clusters contain a single unit. For the k-means clustering

for mean single-unit cue firing, 22 was the first number that produced a cluster with a single unit. Examining the cluster firing means, 21 appeared to capture consistent and unique clusters. However, we concede that selecting any cluster number, no matter how reasonable, involves judgment. In the manuscript we are careful to never claim that there are exactly 21 clusters. Only that the brainstem can be functionally divided into at least 21 clusters.

On which trials the PCA was made (are all trials included in one PCA, and if so, how were the curves for the single experimental conditions computed?). Please add precision in the text or methods.

PCA was performed on mean, single-unit cue firing across all trials. This is now stated on Lines 57-58.

In these clusters, some of the smaller clusters look quite similar in their firing z-score. For example, cluster 1+5, cluster 2+8 or cluster 6+7. There might have been over clustering of the data? Please explain and discuss how baseline changes are likely to change the outcome of the z-score.

Your point is taken about the similarity of clusters. (Note that the identities changed following safety firing correction) Clusters 1+5 and 2+6 appeared to be similar in terms of population firing. However, comparing these clusters in other aspects of firing show differences. Cluster 1 neurons had significantly stronger firing relationships to the cue subnetwork clusters than did cluster 5. For clusters 2 and 6, cluster 2 had shorter and less variable latencies. This pattern held for many clusters that looked similar in terms of population firing. Despite this similarity, no two clusters are identical when considering all aspects of firing (PC1 information, danger latency, cluster-cluster correlations and unit-cluster correlations).

3) Data (clusters) presented in the Fig 2C are interpreted as reflecting different threat probability. However, none of the clusters shows an important difference in firing towards the uncertain or the safe stimulus. It could be that the rats simply did not understand the association of the 'uncertain' tone to the shock? Please discuss.

Behaviorally, we know that rats understood the difference between the cues. Nose poke suppression differed between each cue pair. The data presented in the Fig 2C reflects low dimensional firing information present across all neurons. Cue presentation was evident in this information, as all cues had positive PC1 weights. Clearly, danger had the most positive weight and was 'treated' differently from uncertainty and safety. Importantly, the PC1 weights for uncertainty and safety are not completely overlapping. They are just very close. This pattern would be expected of a signal for threat probability. After all, shock occurs on 0% of safety trials and only 25% of uncertainty trials. These should be treated more similarly than a danger trial on which 100% of trials end in foot shock. Of course, principal components for firing rates cannot determine whether firing reflects threat probability. This is why we later use linear regression.

4) Clusters (about firing latency to danger) presented in Fig 2D seem to contain two groups of cells with different latencies. Maybe these clusters contain different subclusters? Please discuss.

We wondered about this too. However, we think this is more a product of our analysis approach. To determine latency, we broke down cue firing into 1 ms bins (10,000 bins) then looked for the most abrupt change in firing rate slope between adjacent bins. For phasically responsive neurons this worked well, identifying the first deflection from baseline. However, for tonically responsive neurons or very minimally responsive neurons, none of these deflections was larger than the deflection at the very end of cue, which compared the final value to zero. Continuing to look beyond cue offset was not fair as danger trials would always end in shock presentation, which would reveal firing for a totally different reason. Thus, single units with values of 10,000 are those that did not show abrupt deflection during cue presentation.

5) Figure 2E show cue firing correlation between clusters. Clusters 1-8 are shown to have very high correlation. But if there was overclustering in these clusters, the correlation would only indicate that some clusters belong together. How did you checked for overclustering?

Clusters 1-8 indeed have high correlation coefficients. In theory, this could indicate that they are of the same functional type, and we have overclustered. However, these clusters differed in other ways. For example, above it was noted that clusters 1+5, and 2+6 showed somewhat similar mean cue firing patterns. However, when firing latency was compared, differences emerged. So is this case when examining correlated firing. Cluster 1 was superior to cluster 5 in both cluster-cluster and unit-cluster correlations (Fig. 2E). Clusters 2 and 6 show similar cluster x cluster correlations, but cluster 2 neurons show stronger unit x cluster correlations.

We feel that when considering all aspects of firing (mean cue firing, PC1 contribution, danger firing latency, cluster x cluster firing correlations and unit x cluster firing correlations) 21 clusters holds up well. Still, we acknowledge and carefully state that the brainstem does contain exactly 21 functional neuronal types, but at least 21 types.

6) The clustering (Fig 2G) might be explained by the firing onset corresponding to the location of the most cells in the descending pathways from cranial to caudal. Indeed, the danger firing latency in 2D might be dependent on the location of the cells in the brainstem. In figure 2G the clusters that contained more cranial neurons had shorter firing latencies (e.g., cluster 1, cluster 10). Please discuss.

We agree. This was part of the rationale for describing the superior colliculus and periaqueductal gray as hubs for the cue subnetwork. Though not only did cluster 1 neurons have shorter firing latencies, but they also contributed more greatly to PC1 firing information the activity of each individual neuron better correlated to the cue subnetwork. We describe these combined features on Lines 79-82, then relate this to anatomy on Lines 83-85.

7) The authors tried to explain the firing of the single clusters over time in Fig 3 by doing a linear model, in which the threat probability and the behaviour were used as regressors. Please add methodology details for this model. Do the authors validate if the different firing rates in the different threat probability conditions are not representing the reaction to the different tones that were played in the different conditions?

Linear regression is described in the supplemental methods. However, we understand that descriptions alone can make it difficult to determine what exactly regression is doing. To better describe our regression approach, we provide an example regression from the manuscript. In this example, linear regression for Z firing of cluster 1 neurons is being performed to determine whether firing is better explained by threat probability or behaviour. The 16 session trials are organized by type (danger vs. uncertainty vs. safety) and mean Z firing is determined for each individual trial. The matlab regress function requires a constant '1' be used in linear regression. The two regressors are the shock probability specific to the cue (1, 0.25 or 0) and the mean suppression ratio observed to each cue. The output of linear regression is a beta coefficient providing the direction and magnitude of the relationship between each regressor and firing. In this example, both regressors are positive, indicating a positive relationship with firing, but the threat probability regressor is of greater magnitude. Thus, cluster 1 firing in the first 1s of cue presentation is best captured by threat probability.

Linear regression example

Cue cluster 1, first 1 s of cue presentation

Trial #	Trial type	Z firing	Constant	Threat probability	Behaviour
1	danger	1.49	1	1	0.98
2	danger	1.89	1	1	0.85
3	danger	2.42	1	1	0.70
4	danger	2.04	1	1	0.85
5	uncertainty	1.16	1	0.25	0.45
6	uncertainty	1.20	1	0.25	0.79
7	uncertainty	1.32	1	0.25	0.79
8	uncertainty	1.63	1	0.25	0.77
9	uncertainty	1.02	1	0.25	0.64
10	uncertainty	1.18	1	0.25	0.71
11	uncertainty	1.19	1	0.25	0.55
12	uncertainty	0.96	1	0.25	0.48
13	safety	0.77	1	0	0.46
14	safety	0.93	1	0	0.41
15	safety	0.57	1	0	0.23
16	safety	0.52	1	0	0.30
Beta coefficient				0.97	0.49

8) Some of the clusters in Fig 3A seem to be correlating with both threat probability and behaviour regressors. Even if the curves representing the regressors are not identical, they seem to be mirrored around the x axis in some clusters (k6,12,19,20). Why you state that the firing that is explained by behaviour should be not affected by the threat probability?

This is correct. Single clusters can show signaling of both threat probability and behavior. Further there is considerable diversity in both what is signaled: behaviour, probability, or mixed; as well as the temporal properties of signaling. We have made sure there is no mention of exclusive signaling and we point out signaling variety across clusters [Lines 90-94].

9) In Figure 3D, the larger network is disabled and the small one still intact. The beta for the cue firing is smaller in the PC1, which means that the beta for the cue firing is also dependent on the larger network. In the text, the authors state that cue firing would be entirely dependent on the small network. Why do you think then that it was 'entirely' dependent?

We agree that 'entirely' dependent is an overstatement. We have removed this language and instead state that behaviour signaling depended more on the supranetwork, whereas threat probably signaling depended more on the cue subnetwork [Lines 111-117].

10) Brainstem location of the neurons in the larger network containing cluster 9 to 21 and of the smaller cluster is presented in Figure 3E. It looks like the IPAG is more present in the smaller network. In the text, the authors state this was a proof that there is a more dorsal network that computes fast threat probability and a more ventral network that computes behavioural reaction and slow threat probability. Could you be more precise? This statement is too large. First, it is known that there is behaviour computed in both ventrolateral and dorsal PAG. Second, the higher contribution of IPAG neurons in the smaller network could also be due to the location of the neuropixel probe in the brainstem. In Figure 1A it looks as if the Neuropixel was implanted more lateral, touching the lateral PAG directly, but not the ventral part of the brainstem. Therefore, the neurons recorded from the IPAG were closer to the probe and maybe easier to identify in the data than the ones in the ventral areas. Therefore, they might be overrepresented in the network. Thus, is it valid to draw a conclusion of the location of a network on this?

We agree. Further, correcting the safety firing somewhat shifted the location of the clusters contributing to the cue subnetwork and cue supranetwork. It is now the case that anatomical biases were only apparent for the cue subnetwork, and they were subtle [Lines 118-121].

In regards to the higher concentration of IPAG neurons in the subnetwork, we do not think this is due to 'higher' location on the probe. The highest probe locations were actually in retrosplenial and visual cortex. Although we did not collect activity from those portions of the probe because they were not the focus of our study. Further, the region from which we obtained the most neurons was the paramedian raphe, one of the most ventral regions we recorded from. On top of this, when we map out the electrode path through the brain we start by identifying the tip – the most ventral location – then the top – the most dorsal location. Once the extreme are identified we fit a linear path then inspect the entire path to see if areas of low unit yields corresponded to fiber tracts – which they did.

11) In Figure 4 is presented the cluster firing regarding different conditions: Predicted shock, unpredicted shock, unpredicted omission, and predicted omission. The unpredicted omission

occurs in the trial condition where the rat never experiences a shock following the tone. Therefore, the rat did not associate the tone to a shock in first place. If there is no expectation of a shock, how can the shock be omitted? Could it be that the reactions is to the 'safety' condition rather as a pure reaction to the tone as a negative prediction error? Please precise from which trial the 'unpredicted foot shock' originates. If the unpredicted foot shock is the foot shock from the uncertainty condition, it is not a completely unpredicted foot shock. Therefore, the positive predication error signal might be not pure in this condition. Please discuss. In Figure 4B the duration was long, 10s (after the offset). Did the authors check if the firing of the clusters could also be related to motion?

We are careful not to use the term 'unpredicted' anywhere in the manuscript. Instead we use 'surprising'. The uncertainty predicts both shock and omission. However, when the uncertainty cue is presented, the rat does not which trial type is occurring. This is because the order of uncertainty shock and uncertainty omission trials is randomly determined every session. Receipt of foot shock is 'surprising' because there was a 75% chance omission would occur. Receipt of omission is also 'surprising', though less so, because there was a 25% chance shock would occur. This is in contrast to 'predicted' shock on danger trials. Every time the rat hears the danger cue, it receives a foot shock. The shock is fully predicted. The same logic applies to the safety cue. Every time the rat hears the safety cue, it never receives a foot shock. The absence of shock is fully predicted. So we agree that foot shock presentation on uncertainty trials is not 'unpredicted' and we never make that claim. Foot shock presentation on uncertainty trials is 'surprising' because shock omission is also possible.

We address motion below.

12) Linear regression model to explain the weights of either the shock or a prediction error in the clusters firing is presented in Figure 5. Could the authors add a regressor estimating the impact of motion? (to compare with the reaction to the shock?)

Our Neuropixels recording rig is not equipped to quantify locomotion. However, a recent study from a talented graduate student in the lab (Amanda Chu) set out to construct complete temporal ethograms of behavior during our fear discrimination procedure. In brief, she trained 24 rats (12 females) in our danger, uncertainty, safety discrimination; recorded and hand-scored behavior frames prior to and following cue presentation. Nine total behaviors were quantified but here I am showing locomotion. Locomote was defined as propelling the body forward by movement of the legs. The line graph on the left shows mean \pm SEM % locomote behavior. We observe danger-specific increases in locomotion, however these increases are only apparent towards the end of cue presentation. The bar graph on the right shows cue mean (danger, red; uncertainty, purple; and safety, blue) and the points are individual rats. During late cue presentation there was greater locomotion to danger compared to baseline, as well as to danger compared to safety.

Assuming the rats in this study showed the same pattern, late cue firing could in part reflect (loco)motion. Indeed, cue cluster k15 only increases firing during late danger presentation – perhaps signaling locomotion. However, we have many clusters that preferentially fire during early cue presentation (k1, k3, k5, k12, etc.). The activity of these clusters cannot reflect locomotion.

13) It is not clear how the authors reached the conclusion, that the brainstem is constructing threat probability. Indeed, in the figure 1, there seems to be no significant firing difference to uncertain foot shock firing and safety firing. Could it be that the rats did not understand the association between the insecure foot shock and the tone? Even if the signalling would represent threat probability signalling, it is not proof for the construction of the signalling in the brainstem (the signals could be simply conducted by the brainstem)

We see how this was interpreted in the original manuscript. Once we corrected the safety averaging error it was clear that differential firing to safety and uncertainty is just as robust as differential firing to danger and uncertainty.

REVIEWER COMMENTS

Reviewer #1 (Remarks to the Author):

The authors have addressed the major concerns raised in the initial review. Importantly, they have also revised their analyses to account for a bug in their code, which has primarily affected the responses during the safety trials. This has not changed the overall conclusions of the manuscript. In terms of my specific comments, I appreciate the responses offered by the authors and the clarifications that were made.

Reviewer #2 (Remarks to the Author):

The authors have submitted a revised version of their manuscript on threat probability, fear behaviour and aversive prediction error in the brainstem. The authors have adequately addressed part of my concerns. However, some important points still need to be addressed.

From the first to the second version of the paper, the clustering was changed drastically! The reason for this does not seem to be adding more neurons to the analysis, as the numbers of animals and neurons have remained equal. The number of the overall clusters remains the same, but the sizes and response shapes of the single clusters varied a lot, as did the PC1 contribution in 2C. I find this quite curious and would be interested to find out which modifications were carried out (in detail) from the 1st to the 2nd version.

Fig 2 E. After reclustering, clusters 6-9 and clusters 1-2 seem very correlated to each other. Maybe you could adjust the color scale of 2E and show R2 values above 0.5 to make differences between clusters more visible.

Fig 4. The clusters described in these figures have also been reclustered since the first version of the paper. How were they reclustered and why?

Fig 5. Here, there was also some reclustering made. The results are opposite to what was presented in the first article in some cases (5B for example, here, the error now is the main component of PC 1 and not the shock.) Why and how?

The authors are using 'signals for sensory shock' and 'prediction error signalling' for the linear regression. It is not clear why here in figure 5, they chose regression to 'signals for sensory shock' instead of 'fear behaviour' as for the network analysis in figure 3.

Please add a paragraph that explains why the single units are supposed to perceive the sensory shock (input) (fig. 5) although the network is supposed to be representing fear behaviour (output) (fig.3).

The authors are referring to positive prediction error even though there was only unsigned prediction error discovered in this study (line 219). Please correct.

Reviewer #1

The authors have addressed the major concerns raised in the initial review. Importantly, they have also revised their analyses to account for a bug in their code, which has primarily affected the responses during the safety trials. This has not changed the overall conclusions of the manuscript. In terms of my specific comments, I appreciate the responses offered by the authors and the clarifications that were made.

Thank you very much for your feedback. The manuscript is much improved for it.

Reviewer #2

The authors have submitted a revised version of their manuscript on threat probability, fear behaviour and aversive prediction error in the brainstem. The authors have adequately addressed part of my concerns. However, some important points still need to be addressed.

We appreciate your feedback. Many of the concerns below stem from the changed data between the original and revised manuscript. Our revision included a detailed explanation for these changes. We are including that explanation at the end of this reply to make sure the reason for the changed data is clear.

From the first to the second version of the paper, the clustering was changed drastically! The reason for this does not seem to be adding more neurons to the analysis, as the numbers of animals and neurons have remained equal. The number of the overall clusters remains the same, but the sizes and response shapes of the single clusters varied a lot, as did the PC1 contribution in 2C. I find this quite curious and would be interested to find out which modifications were carried out (in detail) from the 1st to the 2nd version.

The reason for the differences in the individual clusters is the corrected firing activity of each neuron during the safety cue between the 1st and 2nd versions. In the first version, firing during the safety cue accidentally included firing during all firing types rather than just the safety cue. The clustering analysis used firing activity of each neuron, during all three cues, meaning that the specific activity patterns of each cluster were altered by correcting firing during the safety cue. As you pointed out however, the total number of clusters that best fit the data remained the same, as does the common feature of many of the clusters in that they are threat sensitive by differentiating between the cues. The altered firing activity during the safety cue, and resulting clustering analysis, therefore also changed the precise contribution to PC1 of each cluster, mostly seen for cluster 2. In version 2, contribution to PC1 is also shown as mean contribution per unit, rather than overall contribution of the cluster. We feel this best reflects PC1 contribution of the neurons within each cluster to PC1 and better accounts for different numbers of neurons within each cluster.

Fig 2 E. After reclustering, clusters 6-9 and clusters 1-2 seem very correlated to each other. Maybe you could adjust the color scale of 2E and show R2 values above 0.5 to make differences between clusters more visible.

Your observations are correct. Clusters 1-2, as well as clusters 6-9 show strong positive correlations with one another. The clear outlier is cluster 3. This was the only cluster to not show ordered cue firing ([danger > uncertainty > safety]). It makes sense this would

not correlate well with most others. Certainly, not all cue clusters are equally correlated. We have edited the manuscript to make this clear. That said, much more is going than just clusters 1-2 and 6-9 being correlated. Cluster 1 strongly correlates with the majority of clusters: 2, 4-8. Cluster 2 strongly correlates with 4, and 6-9. This is almost certainly why clusters 1 and 2 were found to be hubs. This might also suggest an additional level of organization within the cue subnetwork, with clusters 1 and 2 directing smaller networks within. These observations are now clearly stated on Lines 79-83.

Fig 4. The clusters described in these figures have also been reclustered since the first version of the paper. How were they reclustered and why?

All relevant firing analysis were redone after correction of safety firing, including the clustering analysis for the cue and outcome periods. Although the specific firing patterns of the clusters did vary as a result of this analysis, PC1 across all the clusters remained similar with the expectation of activity during the safety cue.

Fig 5. Here, there was also some reclustered made. The results are opposite to what was presented in the first article in some cases (5B for example, here, the error now is the main component of PC 1 and not the shock.) Why and how?

This change in outcome firing activity was also the result of the corrected safety firing and analyses updated to reflect this. In the previous version safety cue firing included activity from all cues, inflating firing activity to the safety cue in the outcome period. Rather, brainstem neurons showed very little firing changes in the safety outcome period, with the corrected analysis revealing opposing firing patterns for shock and prediction error in the way shown in the revised manuscript.

The authors are using 'signals for sensory shock' and 'prediction error signalling' for the linear regression. It is not clear why here in figure 5, they chose regression to 'signals for sensory shock' instead of 'fear behaviour' as for the network analysis in figure 3.

Please add a paragraph that explains why the single units are supposed to perceive the sensory shock (input) (fig. 5) although the network is supposed to be representing fear behaviour (output) (fig.3).

We used linear regression for two separate purposes in this manuscript. For the cue period, we used linear regression to determine whether cluster firing was better described by the foot shock probability associated with each cue (threat probability) or the degree to which nose poke suppression was observed (fear behaviour). For the post-shock period, we used linear regression to determine whether cluster firing was better described by the presence vs. absence of shock (sensory shock) or was preferentially observed to surprising shock delivery (prediction error).

Using threat probability and fear behaviour as regressors for the post-shock period would not be useful, as behaviour during and following foot shock is markedly different from the cue period. Further, once the shock has been received or omitted information about probability is less relevant. Similarly, using sensory shock and prediction error as regressors for the cue period would not be useful. Neither threat probability or fear behaviour signaling could differ on shock delivery and shock omission uncertainty trial types because the rats had no way of knowing which uncertainty trial type they were experiencing.

In the manuscript we do not claim that single units perceive sensory shock. Although we do demonstrate that some single units increase responding equivalently to predicted and surprising shock. We also show there is no relationship between a neurons function during the cue and outcome periods. Therefore, we do not think it is helpful to discuss relationships between single unit shock firing and network level behaviour signaling.

We agree that the relationship between responding at the single unit level and information signaling at the network level was not adequately discussed. We have added a paragraph describing this relationship.

The authors are referring to positive prediction error even though there was only unsigned prediction error discovered in this study (line 219). Please correct.

With the corrected safety firing, we only observe outcome firing consistent with positive prediction error. The statement in the manuscript is correct in light of the corrected safety firing data.

Introduction to the revised manuscript

We are grateful for the thorough and thoughtful critiques from both reviewers. As we revised the manuscript, we meticulously checked our code to verify analyses. During this process we uncovered a simple, yet critical error in our trial averaging. We apologize in advance for this very detailed error description, but feel this is owed and necessary.

Our discrimination sessions consist of 16 trials (4 danger, 2 uncertainty shock, 6 uncertainty omission, and 4 safety). Trial type order is randomized so that every session is unique. To analyze firing and behavior across all sessions, we must first standardize the trial order.

Session trial order		
Session trial #	Trial type	Trial type #
1	danger	1
2	unc omission	1
3	safety	1
4	danger	2
5	unc omission	2
6	unc omission	3
7	unc shock	1
8	safety	2
9	unc omission	4
10	unc omission	5
11	safety	3
12	safety	4
13	unc shock	2
14	unc omission	6
15	danger	3
16	danger	4

Standardized analysis order			
Session trial #	Trial type	Trial type #	Analysis row #
1	danger	1	1
4	danger	2	2
15	danger	3	3
16	danger	4	4
7	unc shock	1	5
13	unc shock	2	6
2	unc omission	1	7
5	unc omission	2	8
6	unc omission	3	9
9	unc omission	4	10
10	unc omission	5	11
14	unc omission	6	12
3	safety	1	13
8	safety	2	14
11	safety	3	15
12	safety	4	16

Above is an example of how this works. The three columns on the left show a hypothetical trial order for one recording session. This is the specific order that rat experienced for that session, and due to random trial selection, no other rat will receive this specific trial order. For each of the 16 session trials we determine the type (of 4 possible) and the number of occurrences for that type. We then rearrange the trial types in a standardized order for analysis. Danger trials are first, uncertainty shock second, uncertainty omission third, and safety last. Trial number is maintained within each trial type. Danger trial 1 is first, trial 2 second, etc. We perform this standardization for every session, which results in a 3D matrix: trial types are rows (X), time bins are columns (Y), and sessions are stacked layers (Z).

Many analyses require us to average all trials of a single type. This is where the error occurred. To find mean danger firing, we average rows 1:4. For uncertainty shock, rows 5:6; uncertainty omission, rows 7:12; and for safety, rows 13:16. However, when we checked our code a '1' was omitted from safety averaging. In error, we averaged rows 3:16 instead of 13:16. What we thought was mean safety firing was actually mean firing for ALL cues, except the first two danger cues. This error profoundly affected reported safety firing.

Having identified the error, we reran all analyses. The correction made obvious impacts on our results. All manuscript figures and supplements have been completely remade to reflect the accurate safety firing data. Correcting safety firing clarified and solidified our findings. Here are some of the biggest changes:

- Brainstem single units showed markedly less firing to safety than either uncertainty or danger during cue presentation (Fig. 2A).
- Functional populations showed discriminative firing that clearly distinguished danger from uncertainty and uncertainty from safety (Fig. 2B).
- Principal components analysis revealed differential cue firing to be the primary low dimensional feature, with PC weights for uncertainty falling almost exactly between danger and safety (Fig. 2C).
- Behaviour (Fig. 3B, PC1) and threat probability (Fig. 3B, PC2) signaling were now more separate low-dimensional firing features with clearer relationships to the supra and subnetworks. The cue subnetwork contributed more greatly to threat probability signaling (Fig. 3C) while the cue supranetwork contributed more greatly to fear behaviour signaling (Fig. 3D).
- Brainstem single units showed almost no firing changes to safety during the outcome period (Fig. 4A).
- Functional populations showed little safety firing during the outcome period, and those that did fire did so in a manner opposing that for uncertainty shock (Fig. 4B).
- Principal components found minimal and negative PC weights for safety (Fig. 4C).

Given this relatively simple error, we were concerned about additional errors. Particularly because many of our analyses are somewhat complex: principal components and regression combined with cluster-specific shuffling. We triple checked all code and found

no additional errors. Looking back, it seems that we were so concerned with nailing down our more complex analyses, that we overlooked one of the simplest errors that could be made.

REVIEWER COMMENTS

Reviewer #2 (Remarks to the Author):

The authors have adequately addressed my concerns in the revision; their manuscript is much improved.

Reviewer #2

The authors have adequately addressed my concerns in the revision; their manuscript is much improved.

Thank you.